# Comparison of mpMRI and ^68^Ga-PSMA-PET/CT in the Assessment of the Primary Tumors in Predominant Low-/Intermediate-Risk Prostate Cancer

**DOI:** 10.3390/diagnostics15111358

**Published:** 2025-05-28

**Authors:** Moritz J. Argow, Sebastian Hupfeld, Simone A. Schenke, Sophie Neumann, Romy Damm, Johanna Vogt, Melis Guer, Jan Wuestemann, Martin Schostak, Frank Fischbach, Michael C. Kreissl

**Affiliations:** 1Devision of Radiology, Department of Radiology and Nuclear Medicine, University Hospital Magdeburg, 39120 Magdeburg, Germany; sebastian.hupfeld@med.ovgu.de (S.H.); neumann@radiologie-sudenburg.de (S.N.); romy.damm@med.ovgu.de (R.D.); frank.fischbach@med.ovgu.de (F.F.); 2Department of Ophthalmology, University Hospital Brandenburg a.d.H., Medical University Brandenburg Theodor Fontane, 14770 Brandenburg an der Havel, Germany; 3Division of Nuclear Medicine, Department of Radiology and Nuclear Medicine, University Hospital Magdeburg, 39120 Magdeburg, Germany; simoneschenke@web.de (S.A.S.); jan.wuestemann@med.ovgu.de (J.W.); 4Practice Schenke, Gießener Straße 37, 35457 Lollar, Germany; 5Department of Nuclear Medicine, Justus Liebig University, 35392 Gießen, Germany; 6Department for Urology, Uro-Oncology, Robot-Assisted and Focal Therapy, Magdeburg University Hospital, Magdeburg Otto von Guericke University, 39016 Magdeburg, Germany; johanna.vogt@med.ovgu.de (J.V.); melisgurmd@gmail.com (M.G.); martin.schostak@med.ovgu.de (M.S.); 7Arbeitskreis für Fokale und Mikrotherapie der Akademie der Deutschen Urologie, 39018 Magdeburg, Germany; 8LOGICURO, 10117 Berlin, Germany; 9Research Campus STIMULATE, Magdeburg University Hospital, Magdeburg Otto von Guericke University, 39016 Magdeburg, Germany

**Keywords:** mpMRI, PSMA-PET/CT, localized prostate cancer, biopsy, localization, focal HDR brachytherapy

## Abstract

While multi-parametric magnetic resonance imaging (mpMRI) is known to be a specific and reliable modality for the diagnosis of non-metastatic prostate cancer (PC), positron emission tomography (PET) using ^68^Ga labeled ligands targeting the prostate-specific membrane antigen (PSMA) is known for its reliable detection of prostate cancer, being the most sensitive modality for the assessment of the extra-prostatic extension of the disease and the establishment of a diagnosis, even before biopsy. **Background/Objectives**: Here, we compared these modalities in regards to the localization of intraprostatic cancer lesions prior to local HDR brachytherapy. **Methods**: A cohort of 27 patients received both mpMRI and PSMA-PET/CT. Based on 24 intraprostatic segments, two readers each scored the risk of tumor-like alteration in each imaging modality. The detectability was evaluated using receiver operating characteristic (ROC) analysis. The histopathological findings from biopsy were used as the gold standard in each segment. In addition, we applied a patient-based “congruence” concept to quantify the interobserver and intermodality agreement. **Results**: For the ROC analysis, we included 447 segments (19 patients), with their respective histological references. The two readers of the MRI reached an AUC of 0.770 and 0.781, respectively, with no significant difference (*p* = 0.75). The PET/CT readers reached an AUC of 0.684 and 0.608, respectively, with a significant difference (*p* < 0.001). The segment-wise intermodality comparison showed a significant superiority of MRI (AUC = 0.815) compared to PET/CT (AUC = 0.690) (*p* = 0.006). Via a patient-based analysis, a superiority of MRI in terms of relative agreement with the biopsy result was observed (*n* = 19 patients). We found congruence scores of 83% (MRI) and 76% (PET/CT, *p* = 0.034), respectively. Using an adjusted “near total agreement” score (adjacent segments with positive scores of 4 or 5 counted as congruent), we found an increase in the agreement, with a score of 96.5% for MRI and 92.7% for PET/CT, with significant difference (*p* = 0.024). **Conclusions**: This study suggests that in a small collective of low-/intermediate risk prostate cancer, mpMRI is superior for the detection of intraprostatic lesions as compared to PSMA-PET/CT. We also found a higher relative agreement between MRI and biopsy as compared to that for PET/CT. However, further studies including a larger number of patients and readers are necessary to draw solid conclusions.

## 1. Introduction

Magnetic resonance imaging (MRI) is an important tool for diagnosis and treatment planning in prostate cancer (PC) [1,2]. Multi-parametric MRI (mpMRI), including dynamic contrast enhanced (DCE) and diffusion weighted imaging (DWI), alongside the common anatomic imaging established as routine tomography for patients with suspected PC. In order to render the reporting more consistent across centers, PI-RADS criteria were introduced [3]. Large trials like the PROMIS study and the PRECISION study led to the conclusion that mpMRI, performed prior to biopsy, not only reduces the number of unnecessary biopsies, but also improves the detection rate of clinically significant prostate cancer [4]. Risk classification by initial MRI, in combination with targeted biopsy of suspicious lesions, proved to be superior to conventional biopsy [5].

PSMA-PET/CT is commonly used in the initial staging of high-risk prostate cancer and is also very well established as a restaging tool in biochemical relapse [6]. Following the German guidelines, the PSMA-PET/CT is not primarily used in intraprostatic localization but to tailor treatment in high-risk PC [3]. Studies like the PRIMARY study demonstrated the potential of PSMA-PET/CT as compared to that for mpMRI and PI-RADS score for determining whether a clinically significant prostate cancer is present [2,7].

In recent years, alternative local prostate therapies have come into focus. High intensity focused ultrasound, laser (for example, the TOOKAD laser), or HDR brachytherapy have been established. By only treating the prostate tumor locally, side effects may be reduced [8,9]. Currently, various clinical trials are ongoing in this field. However, highly sensitive and specific imaging is needed to precisely locate the intraprostatic cancer lesion [3,10].

We aimed to compare the two modalities, mpMRI and ^68^Ga-PSMA-PET/CT, and assess specificity, sensitivity, and congruence with respect to the localization of the intraprostatic malignant lesion (biopsy: ground truth) prior to focal HDR brachytherapy.

## 2. Materials and Methods

### 2.1. Patients

In this study, we compared the sensitivity, specificity, and congruence of mpMRI and ^68^Ga-PSMA-PET/CT in a cohort of 27 patients who were enrolled in a prospective study assessing the efficacy of HDR brachytherapy [11]. The mean age of the patients was 72 years, and the median PSA-level was 9.64 ng/mL (details in Table A5)

Exclusion criteria were previously known extraprostatic tumor manifestations, multiple suspicious intraprostatic lesions (PIRADS 4/5), and contraindications against MRI (i.e., claustrophobia, MR-incompatible implants).

Inclusion criteria were subject’s age between 18 and 80 years, histopathologically confirmed diagnosis of PC, and a Gleason score of 7b or lower.

A total number of *n* = 27 patients provided their informed consent. Documentation and reporting of mpMRI, as well as PET/CT findings, were performed according to standard clinical procedures. All patient examinations were in accordance with the guidelines of the Declaration of Helsinki as of 2013 [12]. The acquisition of subject’s data was part of a prospective study investigating prostate radiotherapy with HDR brachytherapy, supported by interventional MRI, after a positive vote of the local ethics commission (RAD 311; ethics approval: 109/16). We performed a post hoc analysis of the imaging data collected in the context of this prospective trial.

In addition, as a ground truth, histopathologic diagnosis with the use of a systematic biopsy was applied for a part of the patient group by obtaining 10 to 12 biopsy cores under transrectal ultrasound control. Additionally, suspicious areas noted in MRI-imaging were also targeted for biopsy (19 patients with histological ground truth). Using Philips UroNav-software (https://www.usa.philips.com/healthcare/product/HC784026/uronav-mrultrasound-guided-fusion-biopsy-system (accessed on 24 May 2025)), we were able to merge ultrasound and mpMRI sequences in real time [13]. This is implemented as the standard for performing a fusion biopsy. DynaCAD-software (https://www.usa.philips.com/healthcare/product/HC784029/dynacad-prostate (accessed on 24 May 2025)) made it possible to visualize the biopsy cores, as well as the regions of interest (ROIs) from the mpMRI examination [14]. This facilitated a precise localization of the biopsy core in every intraprostatic segment used for ROC analysis. The biopsy cores were numbered; therefore, the respective histopathological findings of the cores could be assigned to the corresponding segment of the prostate (Figure 1).

### 2.2. Measurements

#### 2.2.1. MRI Protocol

The examinations were performed using a Philips Achieva 3T MRI (Philips Medical Systems DMC GmbH, Hamburg, Germany) device. mpMRI included the following acquisitions, according to PI-RADS v. 2.1 [15]: axial diffusion-weighted imaging (DWI) and axial T1-weighted (T1w) imaging, with coverage of the whole prostate. Subsequently, dynamic contrast-enhanced imaging was performed. Native T2-weighted imaging was achieved with two-dimensional (2D) Turbo-Spin Echo (TSE) in three orthogonal orientations, and T1-weighted (T1w) 2D TSE in the axial orientation. DWI was acquired with four b-values (0, 100, 400, 800) mm^2^/s regarding spectral fat saturation. Additionally, a DWI acquisition with a b-value of 1500 mm^2^/s was performed. DCE was conducted as a 3D acquisition with 20 time points over a duration of 150 s, resulting in a time-point resolution of 7.5 s. While slice thickness was 6.6 mm, the spacing between the slices was 3.3 mm, resulting in a slice overlap. The Philips CLEAR technique was used to correct for inhomogeneous coil sensitivity. For an overview of the important imaging parameters of the individual acquisitions, see Table A1.

#### 2.2.2. PET/CT

PSMA-PET/CT was performed on a commercial whole-body scanner (Siemens Biograph mCT, Siemens Healthineers, Erlangen, Germany) using an institutional standard protocol, starting with a contrast-enhanced venous CT of the thorax and abdomen, followed by a low-dose CT for attenuation correction, plus PET acquisition, which includes six bed positions, with a 3 min acquisition time per bed position. The acquisition was performed after weight-adapted intravenous injection of ^68^Ga-PSMA-11 tracer (activity in MBq = weight in kg × 2.5 ≤ 150 MBq), with a median activity of 145 MBq.

The CTscan parameters are listed in Table A2. PSMA-PET/CT reconstruction included the parameters in Table A3.

#### 2.2.3. Image Analysis

We used a scheme of 24 intraprostatic segments, 2 for the semilunar vesicles and 1 for the urethra. This scheme was also used in the PI-RADS score [16]. A total of 19 patients underwent a prostate biopsy with TRUS (transrectal ultrasound). All of them received a fusion biopsy using UroNav (established by PHILIPS Healthcare). This software allows for the combination of ultrasound und MRI and for the visualization of marked ROI (regions of interest) from the MRI to the ultrasound image [13]. In addition, the DynaCAD software, also developed by PHILIPS Healthcare (Hamburg, Germany), was used to locate the exact position of the biopsy cores in the MRI sequences. This way, it was possible to locate the cores within the 24 intraprostatic segments [14] For each modality, two experienced, blinded readers independently scored every segment from 1 to 5, according to a Likert scale representing the risk classification (ranging from 1—clinically significant cancer is very unlikely to be present, up to 5—clinically significant cancer is very likely to be present). Figure 2 shows an example from each of them.

We also calculated a near total agreement (NTA) score to compare mpMRT and PSMA-PET/CT, as well as the location by biopsy. To exclude a possible bias due to cross-segmental tumor involvement, assessments directly adjacent to a tumor-involved segment (histological ground truth) were also considered congruent if the assessment was positive (4 or 5). Giesel et al. used a similar approach to divide the prostate into segments and also calculated a near total agreement score to include adjacent segments [17].

#### 2.2.4. Statistical Analysis

The sensitivity, specificity, and congruence of each method were delineated histopathologically, with biopsy as the ground truth. Therefore, we used two different methods of analysis. Segment-wise, we performed an ROC analysis to assess the relationship between the sensitivity and specificity of each reader. Moreover, we used the mean value of every single segment per modality to perform another ROC analysis.

To demonstrate differences in congruence, we applied a patient focused approach and decided to summarize values with same clinical statement (1–3 means there is likely no cancer; 4–5 means there is likely cancer). Then, we assessed the percentage agreement between the combined scores of the two readers per modality. To focus specifically on segments affected by cancer, we also applied a near total agreement by counting every positive assessment (4–5) adjacent to an affected segment (verified by biopsy) as congruent, as suggested by Giesel et al. [17]. Figure 3 depicts the patient records and how they were analyzed.

Percentage agreement was reported using the median, given the missing normal distribution.

We performed a Wilcoxon test to evaluate the agreement between the readers’ assessment and a DeLong test to compare ROC curves. The analysis itself and the design of the ROC curves were performed using RStudio (2024.04.2) with ggplot2 (open source).

## 3. Results

For analysis, we included 30 data sets from 27 patients. One patient was excluded because no data were available regarding the randomized reporting of MRI or PET, nor for the biopsy.

There were 30 data sets, as three patients received the same examinations (biopsy, MRI, and PSMA-PET/CT), both before and after treatment, to monitor recurrence. The six data sets from three patients were randomized independently of each other. In addition, only one data set per patient was used for the ROC analysis (19 patients with biopsy ground truth) in comparison to biopsy. The duplication therefore relates purely to the comparison between the readers of the MRI and PET/CT in terms of the assessments of the same image sequences. The data sets can therefore be regarded as independent. From this, we generated 810 prostate segments, 447 of them with histological ground truth. They were split in 373 negative segments and 74 positives, with the following distribution of Gleason scores: 3 + 3 in 46 segments, 3 + 4 in 26 segments, and 4 + 3 in 3 segments.

### 3.1. ROC Analysis

The three figures above depict the AUC values and the corresponding ROC curves. The segment-based analysis included 408 segments from 19 patients (biopsy ground truth). The AUC values for mpMRI were 0.770 and 0.781 (Figure 4), respectively, with no significant difference (*p* = 0.748 DeLong). For the PSMA-PET/CT, we found values of 0.684 and 0.608 (Figure 5), respectively, with a significant difference in the DeLong-Test (*p* < 0.001). The mean value of the AUC was 0.815 for mpMRI and 0.689 for PSMA-PET/CT (Figure 6), respectively, with a significant difference (*p* = 0.006 DeLong). DAUC showed significant differences between the pairs of MRI 1/PET 2 (*p* < 0.001), MRI 2/PET 1 (*p* = 0.0495), and MRI 2/PET 2 (*p* < 0.001). Between MRI 1/PET 1, there was no significant difference calculated (*p* = 0.056).

### 3.2. Congruence

The congruence was determined by combining the two individual assessments per modality into one and comparing them against the biopsy data of 19 patients (histological ground truth) an depicted in Figure 7. Of these, there was a median agreement of 83.3% between the MRI and biopsy results (average 83.2%) and 79.2% between the PET/CT and biopsy (average 76.2%) results, with a significant difference (Wilcoxon *p* = 0.034). By performing an assessment of near total agreement, the congruence increased to 100% at the median for MRI and 95.8% for PET/CT, with a significant difference (Wilcoxon *p* = 0.024).

Additionally, we conducted an interzonal comparison by determining the congruence between the two valuations per modality from 25 patients (two image sequences per modality were available). This resulted in a median congruence of 92.6% between MRI 1 and 2 and 88.9% between PET 1 and 2, with no significant difference between the two modalities (Wilcoxon *p* = 0.051). All important AUC values and congruence values are listed in Table 1.

In two patients, the treatment plan had to be changed due to metastases detected by PSMA-PET/CT, and focal HDR brachytherapy was omitted; in one patient a bone metastasis was found; a second patient was found to show metastatic spread to a lymph node.

## 4. Discussion

The aim of this study was to compare the two modalities, mpMRI and PSMA-PET/CT, in terms of sensitivity and specificity, for the intraprostatic detection of prostate cancer lesions prior to focal HDR brachytherapy. The role of mpMRI from staging, via therapy planning, to post-therapeutic follow up is well established [18]. In contrast, PSMA-PET/CT is currently primarily used for extraprostatic staging in high-risk prostate cancer. So far, there is no recommendation for the intraprostatic detection of prostate cancer [1,19].

The only known comparable studies including histopathological ground truth come from Giesel et al. [17] and Sonni et al. [20]. Giesel et al. selected 10 patients with high risk PC and evaluated them based on an eight-segment model of the prostate. Regarding the main mass of the tumor (assumed to be where the biopsy core with the highest malignant infiltration was found), they came to the conclusion that mpMRI and PSMA-PET/CT were equally capable of locating the tumor. Additionally, 63.5% of suspicious tumor areas observed via MRI could also be verified using PET. Similarly, 80.2% of the areas detected via PET were also identified by MRI. When considering near total agreement, the scores improved to 89.4% and 96.8%, respectively [17]. Furthermore, Sonni et al. used a similar approach by analyzing 74 men using prostate segmentation of 12 segments, with three different blinded readers involved per modality (mpMRI and PSMA-PET/CT), and ROC analysis employed. They found AUC values of 0.73 (MRI) and 0.7 (PET/CT), respectively, with no significant difference. A patient-based analysis revealed almost equal detection rates of 83% (MRI) and 85% (PET/CT) [20].

In contrast to the studies mentioned, our study focused on a more detailed and precise localization by using a 24-segment prostate model. We performed the analysis using a higher number of segments in order to obtain the most precise modality for diagnosis prior to focal HDR brachytherapy. Although the number of patients seemed relatively small in comparison (27 patients, 19 with biopsy, 12 with complete biopsy of all 24 intraprostatic segments), we were able to use 447 segments with histological ground truth to generate valid data.

Accordingly, the ROC analysis show significant differences between mpMRI und PSMA-PET/CT. First, we determined no significant difference between MRI 1 und 2 but a significant difference between PET 1 and 2. It can therefore be stated that there is a higher interobserver variability in the evaluation of the PSMA-PET/CT, at least in a 24-segment model, than in the evaluation of the mpMRI in this cohort (based on ROC analysis). One reason for this may be the lack of a standard reporting system for PSMA-PET/CT (except for the one used in studies like that by Emmett et al. [2] or approaches like PSMA-RADS [21]), different from mpMRI, where PI-RADS has been established for many years [22]. Also, the higher interobserver variability might be attributable to differing expertise in the interpretation of PET imaging (both readers had more than 10 years of experience, but reader 1 displayed a clinical focus on PET). In total, the mpMRI reached higher AUC values of 0.770 (MRI 1), 0.781 (MRI 2), and 0.815 (MRI mv), with clearly optimized curves in contrast to the results for PSMA-PET/CT: 0.684 (PET 1), 0.608 (PET 2), and 0.689 (PET mv). Significant differences were determined between every pair of data (MRI 2—PET 2, MRI 1—PET 2, MRI 2—PET 1, and MRI mv—PET mv), apart from those for MRI 1 and PET 1. Consequently, 4 of 5 comparisons showed a significant better ratio between specificity and sensitivity for mpMRI compared to that for PSMA-PET/CT. Secondly, we estimated the percentage congruence by clinically summarizing the individual values per modality. Based on works like that of Israël et al. [23], we decided to employ a clinical approach by grouping the scores per segment of 1 and 2 (likely no cancer) and 3 to 5 (likely cancer) and counting them as congruent in comparison to the biopsy-proven segments.

We reached congruences of 91.1% between MRI 1/MRI 2 and 87.1% between PET 1/PET 2. The paired Wilcoxon test showed no significant difference (*p* = 0.051). This underlines the fact that the clinical approach reduced the interobserver variability in each modality, in contrast to the results of the ROC analysis. The individual assessments were therefore comparable, even if the absolute figures for congruence between MRI 1 and MRI 2 tended to be higher than the congruence of PET 1 and PET 2 (details in Table A4).

Significantly higher congruences were found between the results for biopsy and MRI (mean value 83.2%) compared to those for biopsy and PET (mean value 76.2%). Therefore, it can be concluded that a significantly higher congruence (percentage agreement) was reached between the mpMRI assessment and biopsy results as compared to between the PSMA-PET/CT and biopsy in a direct comparison in this cohort. The near total agreement also underlines this statement. By also counting positively scored segments next to an affected segment (histological proven) as congruent, the scores increased to 96.5% (mean value MRI) and 92.7% (mean value PET), with a significant priority of mpMRI compared to PSMA-PET/CT in patient-based, percentage agreement (*p* = 0.024).

Local brachytherapy was not used in two patients because the PSMA-PET/CT examination revealed extraprostatic manifestations of the prostate carcinoma. Due to the complete data set and the randomization of the intraprostatic image findings, the data were nevertheless included in the statistical analysis. In patients with this tumor stage/Gleason score, no metastasis would have been expected, but it was revealed by PSMA-PET/CT, which significantly altered the treatment plan. This also emphasizes the diagnostic value of PSMA-PET/CT.

The number of cases might be the most limiting factor in this study, although the fine resolution of the prostate segments led to a high number of segments overall. Due to its inferior spatial resolution, PSMA-PET/CT reflects a disadvantage here compared to the results of mpMRI. Moreover, the number of cancer free segments predominates. This has a limiting effect on patient-based analysis and the general observation outside the patients in this study. The patient collective was pre-selected to perform the focal HDR brachytherapy after (MR-based) imaging diagnosis. Therefore, patients with low- and intermediate risk carcinomas were almost exclusively included. An analysis by Nakai et al. [24] of mpMRI or the PRIMARY study [2] for PSMA-PET/CT reporting showed the difficulties regarding the diagnosis of this type of tumor. This makes the diagnostic accuracy reached in this study even more important. A comparable study in patients with a low Gleason score [2] found a considerably lower detection rate than that in our study. This could be an effect of the patient collective studied, i.e., multifocal suspicious lesions were excluded in our study. On the other hand, different equipment and imaging protocols might also play a role.

In addition, the expression of PSMA increases with dedifferentiation (increase in Gleason score) [25]. This may be one reason for the reduced intraprostatic sensitivity of PSMA-PET/CT in our study. With regards to the increasing expression related to the Gleason grade, it should nevertheless be noted that despite 29 segments with Gleason 7, the number of Gleason 3 segments (46) still predominated.

For technical reasons, the biopsy was planned with the help of the preoperative MRI sequences in order to perform the fusion guided biopsy, which lead to a bias in favor of MRI diagnostics. However, two meta-analyses from Llewellyn et al. [26] and Watts et al. [27] showed no significant difference between cognitive and software fusion biopsy, which relativizes the bias in favor of MRI. In the future, it would be interesting to explore a similar analysis by performing a PET/CT guided biopsy and possible changes in detection rates. Also, the higher number of segments compared to that used in other studies, i.e., 24 segments in comparison to 12, might have introduced a bias against PET due to its lower spatial resolution in comparison to that of MRI.

However, it must be noted that the results can only be transferred to a heterogenous patient collective to a limited extent. Because of the preselected patient group in our study, the results obtained by both imaging modalities cannot be generalized, even when taking into account low- to intermediate-risk tumors.

Due to the small number of cases in this pilot study, the pre-selected patient collective, and the statistical, but only limited, clinically relevant significance, further larger multi-center studies are necessary to clearly differentiate the intraprostatic diagnostic possibilities of the mpMRI and PSMA-PET/CT modalities.

## 5. Conclusions

This study suggests that in a small collective of mainly low-/intermediate-risk prostate cancer patients, mpMRI is superior for the detection of intraprostatic lesions as compared to PSMA-PET/CT, according to the ROC analysis. We found a higher relative agreement between MRI and biopsy as compared with PET/CT and biopsy results via a direct and adjusted near total agreement with a significant difference. However, since the use of PSMA-PET/CT also detected unexpected extraprostatic metastasis in 2 of 27 patients, which led to a modification of treatment, the use of both methods in the clinical setting of a planned HDR brachytherapy appears to make sense. Further studies, with a larger number of patients and more readers, are necessary to draw solid conclusions.

## Figures and Tables

**Figure 1 diagnostics-15-01358-f001:**
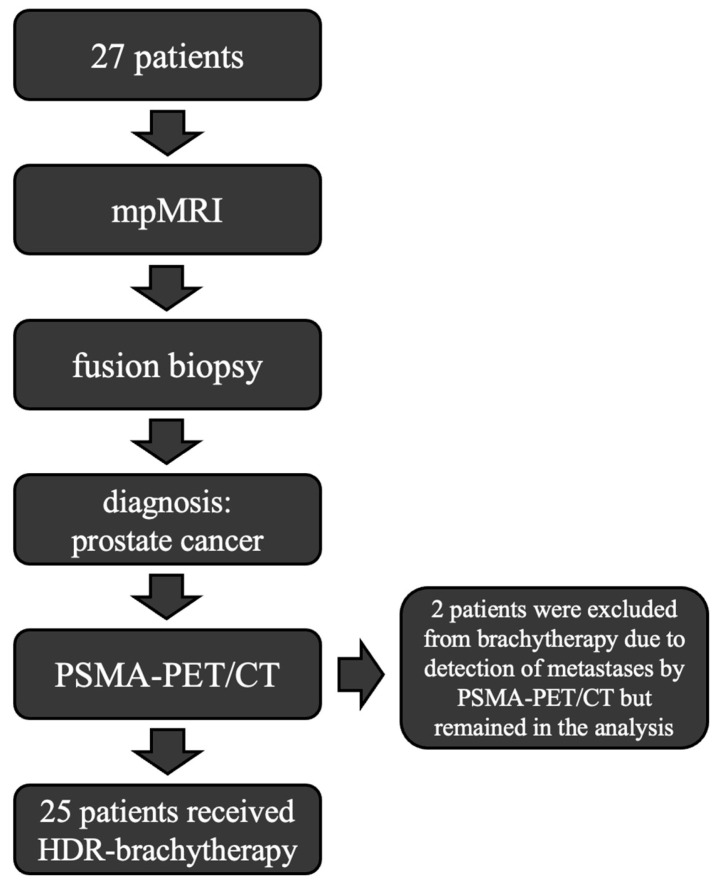
Workflow in this study.

**Figure 2 diagnostics-15-01358-f002:**
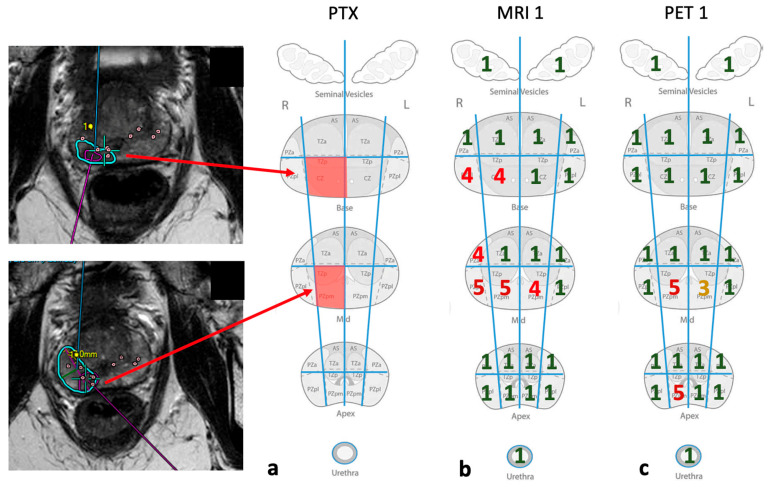
Procedure for the localization of the primary tumor and its assessment by means of the 1–5 scale per segment, indicating the probability for malignancy. The two images on the left show a marked ROI (region of interest), as well as the biopsy cylinders (dots) of the respective assigned level of the prostate (MRI sequence). Using biopsy as the ground truth, the involved segments can be depicted red (PTX) in (**a**). Schemes (**b**,**c**) show the assessments selected by the first reader of each modality (1—dark green, 2—light green, 3—yellow, 4—red, 5—dark red, according to the risk classification mentioned above).

**Figure 3 diagnostics-15-01358-f003:**
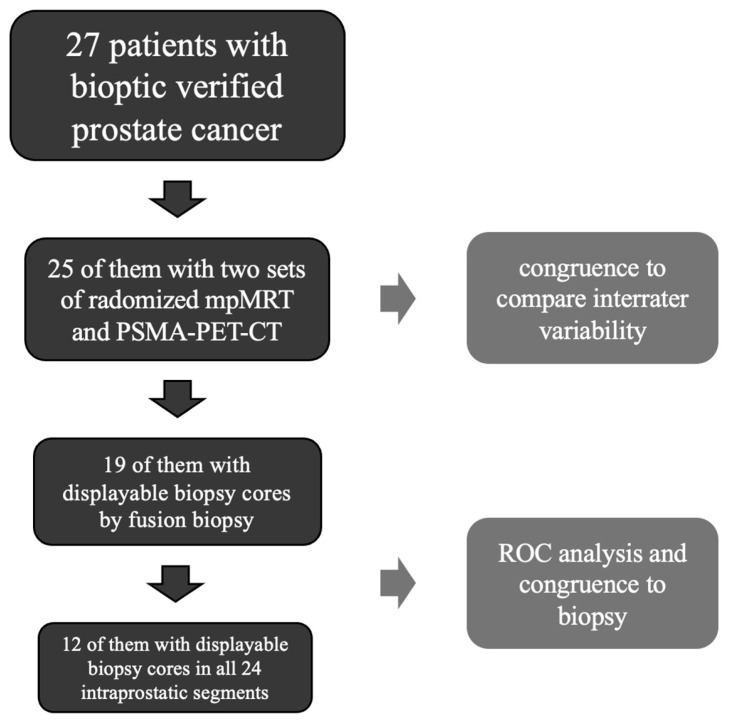
Structure of patient records for each assessment.

**Figure 4 diagnostics-15-01358-f004:**
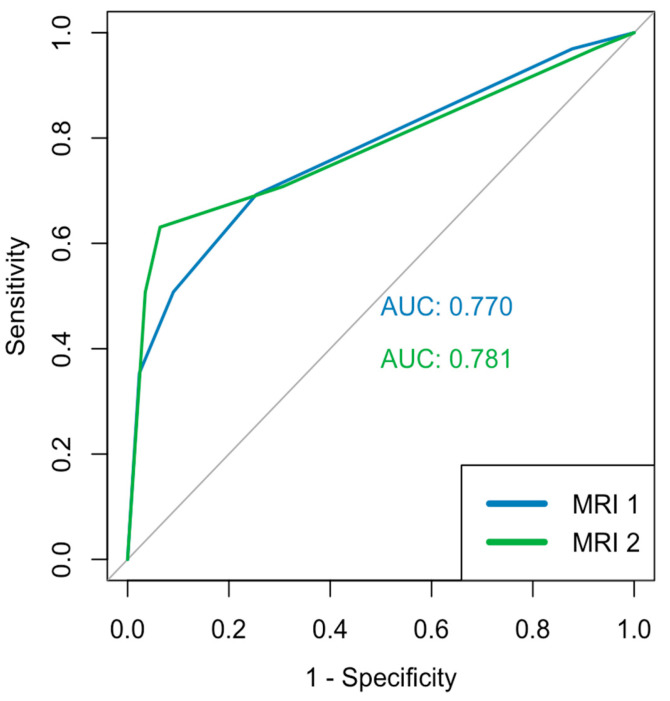
ROC analysis of mpMRI.

**Figure 5 diagnostics-15-01358-f005:**
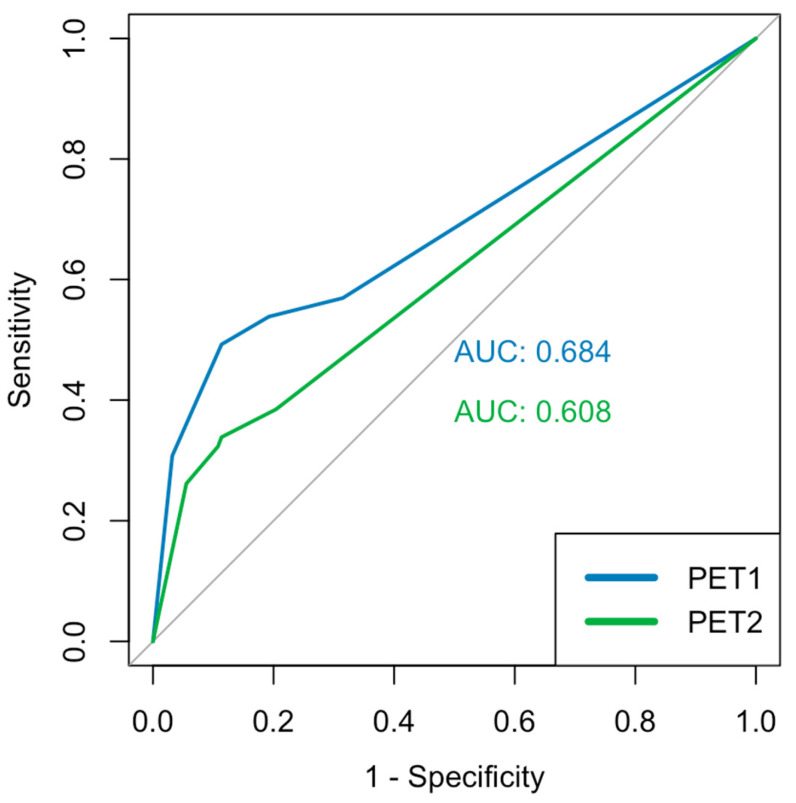
ROC analysis of PSMA-PET/CT.

**Figure 6 diagnostics-15-01358-f006:**
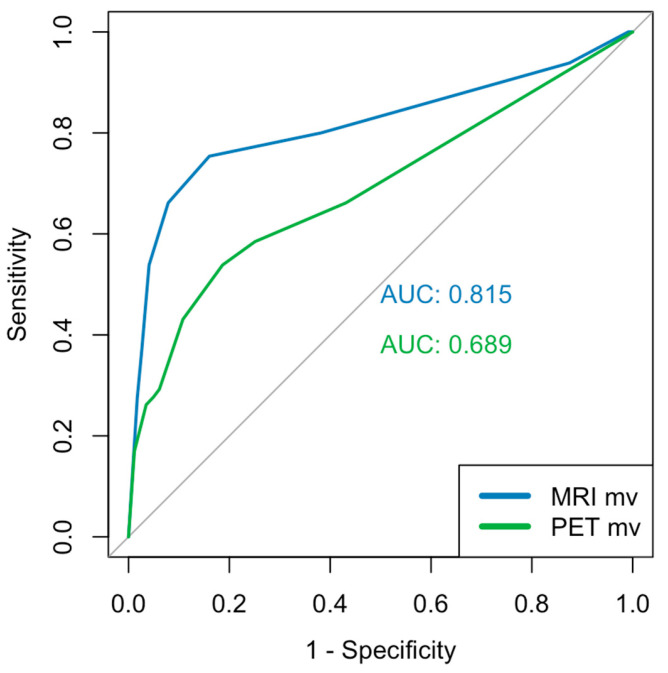
ROC analysis of the mean values per segment of each modality.

**Figure 7 diagnostics-15-01358-f007:**
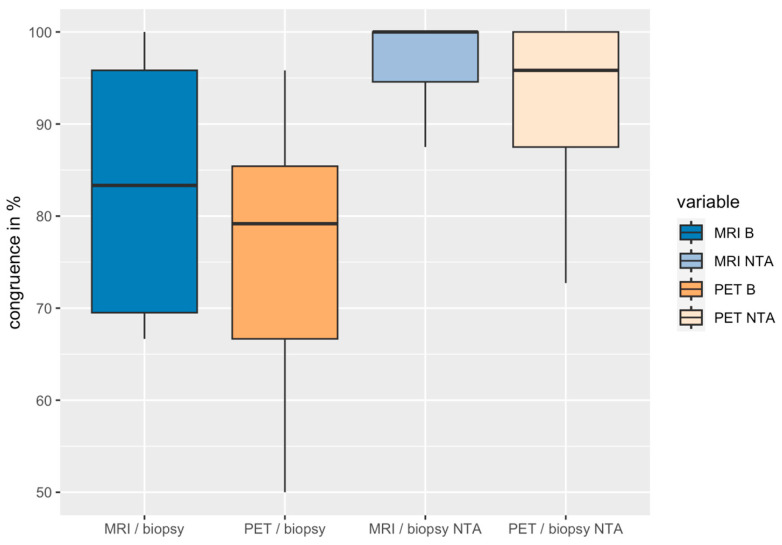
Boxplots of the congruence between biopsy results and those for each modality; assessments for near total agreement (NTA) are shown in lighter colors.

**Table 1 diagnostics-15-01358-t001:** Summarized AUC and congruence values (MRI—magnetic resonance imaging, PET—positron emission tomography—computed tomography, NTA—near total agreement).

	MRI1	MRI2	PET1	PET2	MRI mv	PET mv	MRI/ Biopsy	PET/ Biopsy	MRI/ Biopsy NTA	PET/ Biopsy NTA
AUC	0.77	0.78	0.68	0.61	0.82	0.69				
*p*-value (DeLong)	0.748		<0.001		0.006					
Congruence mv	92.59%		88.89%				83.18%	76.15%	96.54%	92.69%
*p*-value (Wilcoxon)	0.051						0.034		0.024	

## Data Availability

Detailed research data are unavailable due to privacy and ethical reasons.

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
