# Peer review of "Comparison of mpMRI and 68Ga-PSMA-PET/CT in the Assessment of the Primary Tumors in Predominant Low-/Intermediate-Risk Prostate Cancer"

_diagnostics, 2025, doi:10.3390/diagnostics15111358_

Round 1

Reviewer 1 Report

Comments and Suggestions for Authors

Overall Assessment
This study compares the diagnostic performance of mpMRI and PSMA-PET/CT for intraprostatic lesion localization in low-/intermediate-risk prostate cancer patients undergoing focal HDR brachytherapy. The manuscript is well-structured, methodologically sound, and addresses a clinically relevant question. However, the results should be approached carefully due to a few limitations: the small sample size and possible biases. Below are detailed comments.

Given the growing interest in focal therapies, the comparison of mpMRI and PSMA-PET/CT for intraprostatic lesion detection is timely. The 24-segment prostate model enhances spatial precision, critical for brachytherapy planning.

Since the study has a small sample size with only 27 patients (19 with biopsy data), that limits statistical power and generalizability. Also, the predominance of low-risk cases (Gleason ≤3+4) may not reflect higher-risk cohorts. 
The suggestion is that the manuscript might emphasize this as a pilot study and advocate for larger, multi-center validation.
The potential bias: Biopsy targeting was MRI-guided, which may favour MRI's performance. While the authors cite meta-analyses suggesting no difference between cognitive/software fusion, this remains a confounder. The authors might discuss whether standalone PSMA-PET/CT-targeted biopsies could be explored in future work.
The other bias is significant differences between PET readers (AUC 0.684 vs 0.608; *p*<0.001), suggesting subjectivity in PET interpretation. Standardization (e.g., PSMA-RADS) might be proposed to reduce variability.

Also, the spatial resolution limitations of PET's inferior resolution (vs. MRI) may be a disadvantage in a 24-segment model, which is a technical constraint, and whether fewer segments (e.g., 12) would yield different results is debatable.

Consolidation of key findings in a summary table (e.g., AUCs, congruence scores, p values).
Does the 24-segment model explain discrepancies in that there are contrast results with the PRIMARY trial (PSMA-PET/CT vs MRI)? It also should be addressed why PSMA-PET/CT underperformed despite high PSMA expression in Gleason 7 disease.

The abstract should clarify if "447 segments" include all patients or only the 19 with biopsy. Using sensitivity and specificity at optimal thresholds  (not just AUC) is advisable.

Reviewer 2 Report

Comments and Suggestions for Authors

The authors analysed the two modalities mpMRI and Ga-PSMA-Pet/CT in a group of 27 patients. All patients had a histopathologically diagnosed prostate carcinoma. The majority of the 27 patients showed an intermediate risk according to the D'Amico classification, while only one patient had a high risk. The mpMRI and PSMA-PET/CT were compared in terms of specificity, sensitivity and congruence with the localisation of the intraprostatic malignant lesion of the biopsy. PSMA-PET/CT is primarily used for extraprostatic staging only and is not recommended for intraprostatic prostate cancer diagnosis. In this study, the biopsy was performed as a fusion biopsy.

A total of 19 patients showed a positive biopsy result. A comparison of the mpMRI between the two readers showed AUC values of 0.770 and 0.781 (DeLong p = 0.75) and for the PSMA-Pet/CT of 0.684 and 0.608 (DeLong p < 0.001). The combination of both readers showed an advantage for the mpMRI compared to the PSMA-PET/CT (AUC: mpMRI: 0.815, AUC PSMA-PET/CT: 0.690, p = 0.006). In their conclusion, the authors mentioned that mpMRI was superior to PSMA-PET/CT in terms of intraprostatic detection. However, PSMA-PET/CT showed extraprostatic detection of metastases in two patients.

Disadvantages of the study are a small number of patients, the number of examiners, a low number of biopsy-positive prostate segments and the inclusion of patients with a low or intermediate risk according to D'Amico (only one patient with high risk).

The high variance between PET1 and PET2 of 72.17% is striking, while the variance for MRI was 21.79%. There was also a significant difference in the AUC values between PET1 and PET2 (0.684 vs. 0.608, p < 0.001). This could explain the difference between mpMRI and PSMA-PET/CT compared to the intraprostatic lesion of the biopsy. It is unclear why the results of PET1 and PET2 differ so significantly. The authors could explain this in the discussion.
